Full-length transcriptome analysis of Adiantum flabellulatum gametophyte

Cai Zeping 1
Xie Zhenyu 1
Huang Luyao 1
Wang Zixuan 1
Pan Min 1
Yu Xudong 1
Xu Shitao 2
Luo Jiajia luojiajia09skj@163.com 3
1 Key Laboratory of Genetics and Germplasm Innovation of Tropical Special Forest Trees and Ornamental Plants, Ministry of Education, College of Forestry, Hainan University , Haikou , Hainan , China
2 College of Horticulture, Hainan University , Haikou , Hainan , China
3 Tropical Crops Genetic Resources Institute, Chinese Academy of Tropical Agricultural Sciences , Haikou , Hainan , China
Uversky Vladimir
Electronic publication date: 2022 Mar 9
Publication date: 2022
Volume: 10
Electronic Location ID: e13079
Received 2021 Sep 24; Accepted 2022 Feb 16
Copyright: ©2022 Cai et al.
Copyright year: 2022
Copyright holder: Cai et al.
License: This is an open access article distributed under the terms of the Creative Commons Attribution License, which permits unrestricted use, distribution, reproduction and adaptation in any medium and for any purpose provided that it is properly attributed. For attribution, the original author(s), title, publication source (PeerJ) and either DOI or URL of the article must be cited.
License URL: https://creativecommons.org/licenses/by/4.0/

Keywords: Ferns, Three-generation sequencing, Functional annotation, Reference gene set

Funding: Hainan Provincial Natural Science Foundation of China No.319MS017 National Natural Science Foundation of China No.31660229 Hainan University Scientific Research Startup Fund Project No.kyqd1620 National innovation and entrepreneurship training program for college students of Hainan University This work was supported by the Hainan Provincial Natural Science Foundation of China (No.319MS017), the National Natural Science Foundation of China (No.31660229), the Hainan University Scientific Research Startup Fund Project (No.kyqd1620), and the National innovation and entrepreneurship training program for college students of Hainan University. The funders had no role in study design, data collection and analysis, decision to publish, or preparation of the manuscript

==============================
Ferns are important components of plant communities on earth, but their genomes are generally very large, with many redundant genes, making whole genome sequencing of ferns prohibitively expensive and time-consuming. This means there is a significant lack of fern reference genomes, making molecular biology research difficult. The gametophytes of ferns can survive independently, are responsible for sexual reproduction and the feeding of young sporophytes, and play an important role in the alternation of generations. For this study, we selected Adiantum flabellulatum as it has both ornamental and medicinal value and is also an indicator plant of acidic soil. The full-length transcriptome sequencing of its gametophytes was carried out using PacBio three-generation sequencing technology. A total of 354,228 transcripts were obtained, and 231,705 coding sequences (CDSs) were predicted, including 5,749 transcription factors (TFs), 2,214 transcription regulators (TRs) and 4,950 protein kinases (PKs). The transcripts annotated by non-redundant protein sequence database (NR), Kyoto encyclopedia of genes and genomes (KEGG), eukaryotic ortholog groups (KOG), Swissprot, protein family (Pfma), nucleotide sequence database (NT) and gene ontology (GO) were 251,501, 197,474, 193,630, 194,639, 195,956, 113,069 and 197,883, respectively. In addition, 138,995 simple sequence repeats (SSRs) and 111,793 long non-coding RNAs (lncRNAs) were obtained. We selected nine chlorophyll synthase genes for qRT-PCR, and the results showed that the full-length transcript sequences and the annotation information were reliable. This study can provide a reference gene set for subsequent gene expression quantification.

Introduction

Ferns are an ancient group of land plants. Their origins can be traced back to 380 million years ago (Schneider et al., 2004). Ferns are the second largest group of vascular plants (Sessa et al., 2014). At present, more than 11,000 species have been recorded (The Pteridophyte Phylogeny Group, 2016). They are important components of plant communities on Earth.

The genomes of ferns are generally large, with an average size of 12 Gb, and the largest can even reach 148 Gb (Hidalgo et al., 2017a; Hidalgo et al., 2017b). Ceratopteris thalictroides is a model plant of ferns with a genome size of 11.25 Gb, which is more than 80 times the size of the genome of Arabidopsis thaliana (135 Mb; Marchant et al., 2019). Adiantum flabellulatum belongs to the genus Adiantum of Pteridaceae, whose genome size is unknown. The smaller genomes of Adiantum caudatum and Anogramma leptophylla in Pteridaceae are 3.78 Gb and 3.0 Gb, respectively (Kuo & Li, 2019). Because the genes of a fern are highly redundant, whole genome sequencing of ferns is both expensive and time-consuming. The genomes of Azolla filiculoides and Salvinia cucullata are relatively small at 0.75 Gb and 0.26 Gb, respectively (Li et al., 2018). They are currently the only two ferns with whole genome sequences. Due to the lack of reference genomes, the molecular biology research of ferns has somewhat stagnated.

The gametophytes of ferns are very small, but they can survive independently. They are responsible for sexual reproduction, feeding young sporophytes, and are indispensable in the process of alternation of generations. A. flabellulatum is mainly distributed in China, Japan, and Vietnam. Cao, Huang & Wang (2010) observed the gametophyte development process of A. flabellulatum by microscope. Wang et al. (2018) found that plant hormone brassinosteroids (BRs) could regulate the gametophyte development of A. flabellulatum.

For plants without reference genomes, full-length transcripts can effectively provide a reference gene set for gene expression quantification. Jia et al. (2020), for example, used PacBio single-molecule real-time (SMRT) sequencing technology to analyze the full-length transcriptome of Rhododendron lapponicum, and obtained a large number of full-length transcripts, which facilitated gene expression quantification. Wu et al. (2020) analyzed the sequencing data of the full-length transcriptome of endangered species Populus wulianensis, which provided a scientific foundation for an in-depth study of its gene functions. In this study, the full-length transcripts of A. flabellulatum gametophytes were obtained using PacBio three-generation sequencing technology, which can provide a reference gene set for subsequent gene expression quantification and is of great significance to help further the research of gametophyte development in ferns.

Materials & Methods

Plant materials

The spores of A. flabellulatum were collected from the rubber forest in Ma’an mountain, Danzhou City, Hainan Province, China (109.519°E, 19.501°N). The mature spores were sterilized and inoculated in 1/4 MS medium to form young gametophytes after germination. Young gametophytes with diameters of 1–2 mm were selected and transferred into new 1/4 MS medium. These young gametophytes were then placed in illuminations of 0, 10, 500, 1,000 and 10,000 lx, respectively, and cultured at 25 °C for 12 days. Three biological replicates were set in the above 5 illuminations, respectively, with a total of 15 samples. Approximately 200 µg of gametophytes from each sample were selected for RNA extraction.

RNA extraction, Library construction and Sequencing

Total RNA (more than 40 ng/µL) was extracted from the above 15 samples using the CTAB method, and the mRNAs were purified and mixed to construct a cDNA library (Yu et al., 2021). Raw reads were obtained on the PacBio Sequel platform. After reads of insert (ROI) were generated from raw reads, clustering and correction (Chin et al., 2013) were performed using the Quiver algorithm to obtain high-quality full-length consensus reads and then the transcripts were obtained after redundancy removal (using the Cd-hit program, Li & Godzik, 2006). RNA extraction, library construction and sequencing were performed by the Beijing Genomics Institute (BGI).

Coding sequences prediction

The Transdecoder software (v3.0.1, Haas et al., 2013) was used to identify the candidate coding regions in the transcripts. The longest open reading frames were extracted, and the coding sequences (CDSs) were predicted using the SwissProt (http://ftp.ebi.ac.uk/pub/databases/swissprot, Boutet et al., 2007) and protein families (Pfam, http://pfam.xfam.org, Finn et al., 2016) databases.

Simple sequence repeats detection

The MISA software (v1.0) (Thiel et al., 2003) was used to detect the simple sequence repeats (SSRs). The detection parameters were mono-nucleotide, di-nucleotide, tri-nucleotide, tetra-nucleotide, penta-nucleotide and hexa-nucleotide repeats at least 12, 6, 5, 5, 4 and 4 times, respectively. The Primer3 software (v2.2.2) (Untergasser et al., 2012) was used to design primers for the SSRs.

Functional annotation

BLASTN was used for NT (https://blast.ncbi.nlm.nih.gov/Blast) annotations; Blastx (BLAST, v2.2.23, Altschul et al., 1990) or Diamond (v0.8.31) (Buchfink, Xie & Huson, 2015) were used for NR (https://blast.ncbi.nlm.nih.gov/Blast.cgi), KEGG (http://www.genome.jp/kegg), KOG (http://www.ncbi.nlm.nih.gov/KOG) and SwissProt annotations; and Blast2GO (v2.5.0) (Conesa et al., 2005) was used for GO (http://geneontology.org) annotations. After translating CDSs into protein sequences, the protein sequences were uploaded to iTAK (http://itak.feilab.net/cgi-bin/itak/index.cgi) for the prediction of transcription factors (TFs), transcription regulators (TRs) and protein kinases (PKs).

Long non-coding RNAs prediction

The Coding Potential Calculator (CPC) (v0.9-r2) (Kong et al., 2007), txCdsPredict (http://hgdownload.soe.ucsc.edu/admin/jksrc.zip) (Zweig et al., 2008) and the Coding-Non-Coding Index (CNCI) (https://github.com/www-bioinfo-org/CNCI) (Sun et al., 2013) were used to score transcripts other than CDSs, and the scoring thresholds (CPC_threshold = 0, txCdsPredict_threshold = 500, CNCI_threshold = 0) were set to determine long non-coding RNAs (lncRNAs). Pfam was used to predict the protein-coding potential of a transcript. The transcript that could not be predicted in Pfam is potential lncRNA. If the results with three of the above four methods were consistent, the transcript was finally confirmed as lncRNA.

Quantitative real-time polymerase chain reaction

In order to verify the reliability of the full-length transcript sequences and the annotation information, we selected nine chlorophyll synthase genes and used the method applied by Cai et al. (2021) for qRT-PCR (Table S1). Each of the four treatments (0 lx, 10 lx, 500 lx and 10,000 lx) were detected with three biological replicates for a total of 12 samples. Three parallel tests were performed for each sample, and the isoform_55882 (Ubiquitin C) was used as a reference gene. Relative quantification was performed using the 2−ΔΔCt method (Pfaffl, 2001; Cai et al., 2021).

Figure 1 Length distributions of PacBio SMRT sequencing.

(A–D) The length distributions of polymerase reads (A), subreads (B), CCSs (C), and transcripts (D), respectively.

Results

PacBio SMRT sequencing data

In this study, a PacBio ISOSeq library was established, and the full-length transcriptome sequencing was performed on the PacBio Sequel platform. The total base of polymerase reads was 69.87 Gb, and the total number of reads was 923,898. The average length, the longest sequence and the N50 of polymerase reads were 75,625.26 bp, 303,495 bp and 129,782 bp, respectively (Fig. 1A). After removing the adapters, 16,221,787 subreads were obtained, with the total base of 68.72 Gb. The average length, the longest sequence and the N50 of subreads were 4,236.18 bp, 218 714 bp and 4,762 bp, respectively (Fig. 1B).

After subreads from the same circular molecule were combined into circular consensus sequences (CCSs), a total of 809,367 CCSs were obtained. The average length and average quality were 4,934 bp and 0.97, respectively (Fig. 1C). The CCSs were split into full-length, non-full-length, chimeric and non-chimeric sequences according to whether 5′ primer, 3′ primer and ploy(A) tail were detected. There were 2,762,613 full-length non-chimeric (FLNC) sequences, with an average length of 998 bp and an average quality of 0.98 (Fig. S1).

The FLNC sequences from the same copy were grouped into clusters, and the Arrow algorithm (Hong et al., 2020) was used to correct the error. After removing the low-quality sequences, 2,608,705 consensus reads were obtained with an average quality of 0.99 (Fig. S2). After removing redundancy, 354,228 transcripts were finally obtained (Table 1, Fig. 1D). The CCS and transcript data have been uploaded to the NCBI Sequence Read Archive (SRA) database (https://www.ncbi.nlm.nih.gov/sra), and the accession numbers are PRJNA774117 and PRJNA733457, respectively.

CDS prediction

CDS prediction was performed on transcripts, and a total of 231,705 CDSs were obtained. The total length was 194,628,012 bp, and the N50 was 1,056 bp (Table 2). There were 164,535 CDSs with a length of 100–1,000 bp, accounting for 71.0% of the total. There were 57,009 CDSs of 1,000–2,000 bp, 8,722 CDSs of 2,000–3,000 bp and 1,439 CDSs of ≥ 3,000 bp, accounting for 24.6%, 3.8% and 0.6% of the total, respectively (SRA database; Fig. 2A). The CDSs have been uploaded to the SRA database, and the accession number is PRJNA777740.

Table 1 PacBio SMRT sequencing data statistics.

Item	Total reads	Total base (GB)	Max length (bp)	Mean length (bp)	N50 (bp)	
Polymerase reads	923,898	69.87	303,495	75,625.26	129,782	
Subreads	16,221,787	68.72	218,714	4,236.18	4,762	
Circular consensus sequences (CCSs)	809,367	3.99	62,285	4,934	4,521	
Full-length non-chimeric (FLNC) reads	2,762,613	2.76	41,488	998	873	
Consensus reads	2,608,705	2.59	41,488	994	871	
Transcripts	354,228	0.46	41,488	1,308.60	1,667	

Table 2 Quality metrics of predicted CDSs.

Total number	Total length (bp)	N50 (bp)	N90 (bp)	Max length (bp)	Min length (bp)	GC (%)	
231,705	194,628,012	1,056	420	20,595	297	47.89%	

Figure 2 Length distribution of CDSs and number of various SSR motifs.

(A) Number of CDSs in different length distributions; (B) number of various SSR motifs.

SSR detection

Through SSR detection of 354,228 transcripts, 138,995 SSRs were obtained from 81,304 transcripts, with a total length of 3,875,607 bp (Table S2). Of these, 50,482 transcripts contained one SSR locus and 30,822 transcripts contained two or more SSR loci (Table 3). Among SSR repeat types, di-nucleotide repeats (90,741) accounted for the highest proportion (65.28%). The second was the mono-nucleotide repeats (35,856) accounting for 25.8% of total repeats and the third was tri-nucleotide repeats (10,528) accounting for 7.56% of repeats. Tetra-nucleotide, penta-nucleotide and hexa-nucleotide repeats accounted for the minority at 0.40%, 0.25% and 0.69% of repeats, respectively. Forty-one thousand three hundred sixty-eight SSRs were present in compound formation. SSRs with six repeat units (19,697, 14 17%) were the most common, followed by seven repeat units (12,575, 9.05%), eight repeat units (9,806, 7.05%) and 12 repeat units (9,272, 6.67%) (Table 4). Among the SSRs with di-nucleotide repeats, AG/CT (68,289) was the most common type, and AT/AT (824) was the least. Among the SSRs with tri-nucleotide repeats, ATC/GAT (2,375) was the most common type and AAT/ATT (51) was the least common (Fig. 2B). Twenty-nine thousand five hundred eighty-nine pairs of primers were designed and screened by Primer3. The primer lengths were 18–28 bp, the GC contents were 20.00–72.22%, the annealing temperatures were 57–63 °C, and the product lengths were 80–276 bp (Table S3).

Table 3 Summary of SSRs identified in the transcripts.

Searching item	Number	
Total number of transcripts examined	354,228	
Total size of examined sequences (bp)	463,542,116	
Total number of identified SSRs	138,995	
Total length of SSRs	3,875,607	
Number of SSR containing sequences	81,304	
Number of sequences containing more than 1 SSR	30,822	
Number of SSRs present in compound formation	41,368	

Table 4 Types of SSR repeat units detected.

Number of repeat units	Mono nucleotide	Di nucleotide	Tri nucleotide	Quad nucleotide	Penta nucleotide	Hexa nucleotide	Total	Percentage (%)	
4	0	0	0	0	291	786	1,077	0.77	
5	0	0	6,278	377	48	133	6,836	4.92	
6	0	17,114	2,412	130	13	28	19,697	14.17	
7	0	11,672	854	37	1	11	12,575	9.05	
8	0	9,247	547	12	0	0	9,806	7.05	
9	0	7,700	223	2	0	0	7,925	5.70	
10	0	6,181	107	0	0	0	6,288	4.52	
11	0	4,946	58	0	0	0	5,004	3.60	
12	4,837	4,411	24	0	0	0	9,272	6.67	
13	3,677	4,014	19	0	0	0	7,710	5.55	
14	2,899	3,546	6	0	0	0	6,451	4.64	
≥15	24,443	21,910	0	0	0	1	46,354	33.35	

Functional annotation

Three hundred fifty-four thousand two hundred twenty-eight transcripts were annotated by seven databases. There were 251,501, 197,474, 193,630, 194,639, 195,956, 113,069 and 197,883 transcripts annotated by the NR, KEGG, KOG, Swissprot, Pfma, NT and GO databases (Table S4), respectively. Most of them were annotated by the NR database, accounting for 71.00%. The NT database annotated the least, accounting for 31.92% (Table 5). Fifty-five thousand seven hundred ninety-four transcripts were annotated by all the seven databases combined, accounting for 15.75% of the transcripts. There were 125,336 transcripts annotated by NR, KEGG, KOG, SwissProt and Pfam databases together, accounting for 35.38% (Fig. 3A).

Table 5 Number of transcripts annotated by seven databases.

Item	Total	NR	KEGG	KOG	Swissprot	Pfma	NT	GO	Intersection	Overrall	
Number	354,228	251,501	197,474	193,630	194,639	195,956	113,069	197,883	55,94	277,198	
Percentage	100%	71.00%	55.75%	54.66%	54.95%	55.32%	31.92%	55.86%	15.75%	78.25%	
Notes.

Note: the “Intersection” expresses the number and proportion of transcripts annotated by 7 databases together; the “Overrall” expresses the number and proportion of transcripts annotated by at least 1 database.

Figure 3 Venn diagram of transcript number distribution annotated by 5 databases and homologous species information in the NR database.

(A) Venn diagram of transcripts annotated by NR, KEGG, KOG, Swissprot and Pfam databases. The numbers in the overlapping areas represent the numbers of shared genes. (B) The proportion of transcripts derived from homologous species and annotated by the NR database.

NR annotation

The proportion of transcripts annotating to homologous species was counted from the NR annotation results. Among them, the proportions of transcripts annotating to Marchantia polymorpha, Selaginella moellendorffii, Physcomitrella patens and Picea sitchensis were 59.31%, 10.98%, 9.45% and 8.70%, respectively, and the proportion of transcripts annotating to other known species was 11.56% (Table S5; Fig. 3B).

KEGG annotation

A total of 197,474 transcripts were annotated by the KEGG database, of which 112,839 were annotated to 137 pathways (Table S6). The KEGG pathways were divided into five categories: “cell processing,” “environmental information processing,” “genetic information processing,” “metabolism” and “organismal system.” For “metabolism,” “global and overview maps” had the greatest number of transcripts (55,321), accounting for 28.01%, followed by “carbohydrate metabolism,” which had 21,432 transcripts, accounting for 10.85% (Fig. 4A).

Figure 4 KEGG pathway analysis, KOG functional classification, and GO analysis of transcripts.

(A–C) KEGG pathway classification (A), KOG functional classification (B), and GO functional classification (C) of transcripts. Note A: 1: Transport and catabolism; 2: Signal transduction; 3: Membrane transport; 4: Translation; 5: Folding, sorting and degradation; 6: Transcription; 7: Replication and repair; 8: Global and overview maps; 9: Carbohydrate metabolism; 10: Energy metabolism; 11: Amino acid metabolism; 12: Lipid metabolism; 13: Biosynthesis of other secondary metabolites ; 14: Metabolism of cofactors and vitamins; 15: Metabolism of other amino acids; 16: Metabolism of terpenoids and polyketides; 17: Nucleotide metabolism; 18: Glycan biosynthesis and metabolism; 19: Environmental adaptation. Note B: 1: General function prediction only; 2: Signal transduction mechanisms; 3: Posttranslational modification, protein turnover, chaperones; 4: Function unknown; 5: Carbohydrate transport and metabolism; 6: Energy production and conversion; 7: Transcription; 8: Translation, ribosomal structure and biogenesis; 9: Cell wall/membrane/envelope biogenesis; 10: RNA processing and modification; 11: Lipid transport and metabolism; 12: Cytoskeleton; 13: Intracellular trafficking, secretion, and vesicular transport; 14: Amino acid transport and metabolism; 15: Secondary metabolites biosynthesis, transport and catabolism; 16: Inorganic ion transport and metabolism; 17: Defense mechanisms; 18: Cell cycle control, cell division, chromosome partitioning; 19: Chromatin structure and dynamics; 20: Replication, recombination and repair; 21: Extracellular structures; 22: Coenzyme transport and metabolism; 23: Nucleotide transport and metabolism; 24: Nuclear structure; 25: Cell motility.Note C: 1: Cellular process; 2: Metabolic process; 3: Biological regulation; 4: Regulation of biological process; 5: Response to stimulus; 6: Localization; 7: Signaling; 8: Developmental process; 9: Multicellular organismal process; 10: Positive regulation of biological process; 11: Negative regulation of biological process; 12: Reproduction; 13: Reproductive process; 14: Interspecies interaction between organisms; 15: Growth; 16: Multi-organism process; 17: Immune system process; 18: Detoxification; 19: Rhythmic process; 20: Carbon utilization; 21: Locomotion; 22: Nitrogen utilization; 23: Biological adhesion; 24: Intraspecies interaction between organisms; 25: Pigmentation; 26: Carbohydrate utilization; 27: Cellular anatomical entity; 28: Intracellular; 29: Protein-containing complex; 30: Binding; 31: Catalytic activity; 32: Transporter activity; 33: Structural molecule activity; 34: Transcription regulator activity; 35: Molecular function regulator; 36: Translation regulator activity; 37: Antioxidant activity; 38: Molecular transducer activity; 39: Protein folding chaperone; 40: Small molecule sensor activity; 41: Nutrient reservoir activity; 42: Protein tag; 43: Molecular carrier activity; 44: Cargo receptor activity; 45: Toxin activity.

KOG annotation

There were 193,630 transcripts annotated by the KOG database, which could be divided into 25 groups according to KOG functional classification. Among them, the transcripts belonging to “general function prediction only” were the most common with a total of 48,175, accounting for 24.88%. There were 26,065 “signal transduction mechanisms” and 23,151 “posttranslational modifications protein turnover chaperones,” accounting for 13.46% and 11.96%, respectively (Fig. 4B).

GO annotation

A total of 197,883 transcripts were annotated by the GO database. GO domains can be divided into three categories: “biological processes,” “cell components” and “molecular functions.” A large number of transcripts in “biological processes” were mainly involved in the “cellular process” (94,069) and the “metabolic process” (81,073). The category “cell components” mainly consisted of transcripts involved in “cellular anatomical entity” (129,811) and “intracellular” (82,674). The “Cell components” category mainly consisted of transcripts involved in “binding” (95,413) and “catalytic activity” (95,176) (Fig. 4C).

Prediction of TFs, TRs and PKs

TFs play an important role in gene expression regulation. A total of 5,749 CDSs were predicted belonging to 64 TF families. Among them, the predicted sequences of C3H family were the most common (614), followed by bHLH (449), STAT (1) was the least common (Table 6; Table S7; Fig. 5). In addition, 2,214 sequences belonged to 24 TR families. Among them, the SET family had the most sequences (251), followed by the GNAT family (248), and the two families that had the least number of sequences were RB (4) and MED7 (4) (Table 6; Table S8; Fig. 6).

PKs regulate protein activity and amplify signals to induce a cellular response during plant signal transduction (Ardito et al., 2017). A total of 4,950 CDSs were predicted belonging to the PK families. Among them, 833 sequences belonged to leucine rich-repeat receptor-like kinases (LRR-RLKs) families and 746 sequences belonged to receptor-like cytoplasmic kinase (RLCK) families, accounting for 16.83% and 15.07% of the total PKs, respectively (Table 6; Table S9; Fig. 7).

Table 6 Predicted results of transcription factors, transcriptional regulators, and protein kinases.

Item	Total transcripts	Transcription factors	Transcription regulators	Protein kinases	
Number	354,228	5,749	2,214	4,950	
Percentage		1.62%	0.63%	1.39%	

Figure 5 Transcription factor classification.

Figure 6 Transcription regulator classification.

Figure 7 Protein kinase classification.

(A) Protein kinase classification; (B) LRR-RLK classification; (C) RLCK classification.

lncRNAs prediction

106,931, 118,406, 99,672 and 115,438 lncRNAs were predicted using CPC, txCdsPredict, CNCI tools and the pfam database, respectively (Table S10). Among them, 85,487 were predicted using the 4 methods together, and 2,105, 7,652, 1,839 and 14,710 were predicted by three of the four methods alone (Fig. 8). Finally, 111,793 lncRNAs were obtained, accounting for 31.56% and 91.24% of all transcripts and non-CDSs, respectively.

Figure 8 Venn diagram of the lncRNAs numbers predicted by CPC, txCdsPredict, and CNCI software and the Pfam database.

Numbers in the overlapping areas represent the number of shared lncRNAs.

Quantitative real-time polymerase chain reaction

Chlorophyll is an important pigment in plant photosynthesis, and its synthesis enzyme gene expressions are regulated by light (Lee et al., 2005; Masuda et al., 2003; Matsumoto et al., 2004; Okamoto et al., 2020; Tzvetkova-Chevolleau et al., 2007). In order to verify the reliability of the full-length transcript sequences and the annotation information, we selected nine chlorophyll synthase genes according to KEGG annotation for qRT-PCR, including HemA, HemL, HemB, CHLH, CHLD, CRD, DVR, POR and CAO. The results showed that the expressions of the above nine genes increased at first and then decreased with the increase of illumination (Table S11; Fig. 9). It can be seen that the chlorophyll synthase genes in gametophyte of A. flabellulatum are also regulated by light. This also shows that the data of the full-length transcripts has strong reliability.

Figure 9 Quantitative real-time polymerase chain reaction.

According to the KEGG annotation, nine chlorophyll synthase genes were selected for qRT-PCR. The results showed that the expressions of these nine genes first increased and then decreased with increasing illumination.

Discussion

Based on the Illumina HiSeq sequencing platform, the transcriptomes of roots and shoots of Athyrium yokoscense (Ukai et al., 2020), young leaves of Asplenium nidus, young leaves of Asplenium komarovii (Zhang et al., 2019) and sexual reproduction and apogamy of Adiantum reniforme gametophytes (Fu & Chen, 2019) were studied, and 35,681, 89,741, 77,912, 333,352, 264,791 transcripts were obtained, respectively. The sequencing read length of the Illumina HiSeq is short, the mRNAs need to be broken into short fragments before sequencing, and the complete transcripts can be obtained only by splicing. The genomes of ferns are huge, and the genes are highly redundant, so errors inevitably occur in the splicing process. Therefore, in this study, full-length transcriptome sequencing of A. flabellulatum gametocytes was performed using PacBio SMRT technology. This technology has very long average read lengths and can read the gene sequences completely. We believe that this technology is more suitable for species with high gene redundancy, which may be one of the reasons for the large number of transcripts obtained in this study.

SSRs, also known as short tandem repeats (STRs) or microsatellites (Xu, Sun & Li, 2010), play an important role in genetic development and gene expression regulation (Lawson & Zhang, 2006). Existing studies have shown that AG/CT is the dominant motif of di-nucleotide repeats in ferns, dicotyledons and monocotyledons (Jia et al., 2014; Jia et al., 2016; Kumpatla & Mukhopadhyay, 2005; Liu et al., 2016; Morgante, Hanafey & Powell, 2002; Zheng, Zhang & Wu, 2011), which is consistent with the SSR detection results of A. flabellulatum in this study. AT/AT is the second most common di-nucleotide repeat motif in dicotyledons and monocotyledons in other studies (Kumpatla & Mukhopadhyay, 2005; Zheng, Zhang & Wu, 2011), but in this experiment, AT/AT was the least common di-nucleotide repeat motif. CCG/CGG is the largest tri-nucleotide repeat motif in monocotyledons (Morgante, Hanafey & Powell, 2002; Zheng, Zhang & Wu, 2011), but is rare in dicotyledons and ferns, and the number of CCG/CGG in this study only accounted for 0.19% of the total SSRs.

The NR annotation results showed that the dominant species were S. jiangnanensis (10.98%, lycophytes/ Lycopodiopsida), M. polymorpha (11.56%, bryophytes), P. patens (9.45%, bryophytes), and P. sitchensis (8.70%, gymnosperms). A. flabellulatum belongs to Polypodiopsida (true ferns), which confirms that there is a certain relationship between Polypodiopsida and Lycopodiopsida in plant evolution (The Pteridophyte Phylogeny Group, 2016). In the history of plant evolution, ferns have a relatively close relationship with bryophytes and gymnosperms (Wickett et al., 2014), which is also why the transcripts can be annotated to bryophytes and gymnosperms. The NR database annotation results are consistent with the plant genetic relationship, so this study can provide similar reference gene sequences for other ferns.

TFs, TRs and PKs are important components of cell signal transduction, and also important elements in the gene regulatory network. They are widely present in different organisms and participate in various life activities (Zheng et al., 2016). Among about 25,000 genes in the angiosperm model plant A. thaliana, the numbers of TFs and PKs were 2,391 and 1,028, accounting for about 9.56% and 4.11%, respectively (Aglawe et al., 2012). There are about 30,000 genes in Sorghum bicolor, including 2,654 TFs, accounting for about 8.85% of the plant’s genes (Baillo et al., 2019). Liaquat et al. (2021) detected 163,834 transcripts in Schima superba, and obtained 9,423 TFs, accounting for 5.75% of the transcripts. Of the 181,141 transcripts in Huperzia serrata, 3,391 (1.87%) were TFs (Yang et al., 2017). There were 101,448 transcripts of Monachosorum maximowiczii, including 1,130 TFs, accounting for 1.11% of the transcripts (Liu et al., 2016). In this study, a total of 5,749 TFs (1.62%) and 4,950 PKs (1.39%) were predicted from 354,228 full-length transcripts. It can be seen that the proportions of TFs and PKs in the gametophyte transcripts of A. flabellulatum are less than those in angiosperms, which may be related to the general characteristics of ferns or the relatively simple structure of fern gametophytes.

Conclusions

In this study, full-length transcriptome sequencing of A. flabellulatum gametocytes was performed using PacBio SMRT technology. The genomes of ferns are huge, and the genes are highly redundant, so this was a new attempt at obtaining full-length transcripts of ferns using this technology. Moreover, we predicted lncRNAs of ferns for the first time, and finally obtained 111 793 lncRNAs, accounting for 31.56% of all transcripts. This indicates that a large number of lncRNAs exist in ferns. A total of 354,228 transcripts were obtained in this study which can provide a reference gene set for subsequent gene expression quantification.

Supplemental Information

Supplemental Information 1 Information of primers used in qRT-PCR analysis

Click here for additional data file.

Supplemental Information 2 A total of 138 995 SSRs were obtained by SSR detection of transcripts

Click here for additional data file.

Supplemental Information 3 29 589 pairs of SSR primers were designed and screened by Primer3 software

Click here for additional data file.

Supplemental Information 4 Annotation results of 354 228 transcripts in 7 databases

Click here for additional data file.

Supplemental Information 5 Homologous species distribution of A. flabellulatum transcripts annotated in the non-redundant (NR) database

Click here for additional data file.

Supplemental Information 6 112 839 transcripts were annotated to 137 KEGG pathways

Click here for additional data file.

Supplemental Information 7 5 749 TFs were predicted by itak website

Click here for additional data file.

Supplemental Information 8 2 214 TRs were predicted by itak website

Click here for additional data file.

Supplemental Information 9 4 950 PKs were predicted by itak website

Click here for additional data file.

Supplemental Information 10 LncRNAs were predicted by CPC, txCdsPredict, CNCI softwares and Pfam database

Click here for additional data file.

Supplemental Information 11 Processing and analysis of PCR results

Click here for additional data file.

Supplemental Information 12 Length and quality distributions of the A. flabellulatum FLNC reads

(A) Length distributions of FLNC reads; (B) quality distributions of FLNC reads.

Click here for additional data file.

Supplemental Information 13 Length and quality distributions of the A. flabellulatum consensus reads

(A) Length distributions of consensus reads; (B) quality distributions of consensus reads.

Click here for additional data file.

We sincerely thank reviewers and editors of the PeerJ for providing valuable suggestions on this paper.

Additional Information and Declarations

Competing Interests

Author Contributions

DNA Deposition

Data Availability

The authors declare there are no competing interests.

Zeping Cai and Jiajia Luo conceived and designed the experiments, performed the experiments, analyzed the data, prepared figures and/or tables, authored or reviewed drafts of the paper, and approved the final draft.

Zhenyu Xie performed the experiments, analyzed the data, prepared figures and/or tables, authored or reviewed drafts of the paper, and approved the final draft.

Luyao Huang analyzed the data, authored or reviewed drafts of the paper, and approved the final draft.

Zixuan Wang and Min Pan analyzed the data, prepared figures and/or tables, and approved the final draft.

Xudong Yu and Shitao Xu conceived and designed the experiments, authored or reviewed drafts of the paper, and approved the final draft.

The following information was supplied regarding the deposition of DNA sequences:

The full-length transcript data is available in the SRA database: PRJNA733457.

The following information was supplied regarding data availability:

The data is available at NCBI SRA: PRJNA774117, PRJNA733457, PRJNA777740.

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
