# Peer review of "Full-length transcriptome analysis of Adiantum flabellulatum gametophyte"

_PeerJ, doi:10.7717/peerj.13079_

## Round 0.1 · original submission · Major Revisions

Please address the concerns of both reviewers and revise the manuscript accordingly.

Reviewer 1 ·

Basic reporting

The language of this manuscript needs further enhancement. Problem statement in the abstract and Introduction is unclear. Simply stating “The study of gene expression is rarely reported” is not a strong enough reason to conduct full-length transcriptome analysis. I suggest the authors to highlight the potential impact of their study.

Experimental design

Experimental Design
Some parts of the experimental design are unclear. For example:
1. Line 80-82: How many gametophytes were used for RNA extraction?
2. Line 86: were the gametophytes pooled according to illuminations?
Others are quite standard.

Validity of the findings

no comment

Additional comments

The study is a fundamental descriptive study. The authors should have incorporated treatment comparison, e.g. comparison of gametophytes at different illuminations, or different stages, in order to derive more meaningful results. The current manuscript lacks importance and is merely a basic study to provide a reference baseline for future research.

Reviewer 2 ·

Basic reporting

Comments on language and revision: The English language should be improved to ensure that an international audience can clearly understand your text. I suggest you have a colleague who is proficient in English and familiar with the subject matter review your manuscript, or contact a professional editing service. The sections that need particular attention is the abstract and introduction.

Sufficient background and context has been provided in the literature review to understand the aims and rationale of the study.

The study is logically structured and figures and tables are appropriate.

Data: I have looked at the Bioproject and accession and is a bit worried about the file size reported and the data reported in the study. The Bioproject indicate that 113 Mb of 463 542 116 bases has been submitted, while the project indicate that 69 Gb of bases were generated. Perhaps just confirm that these are correlated since I would have expected a few Gb of data. Is this just compressed?

Experimental design

The study lies within the aims and scope of the journal.
The research proposed fills a knowledge gap and the aim of the study is certainly meaningful even if no specific research question was posed.
The experimental procedures are standard for this kind of transcriptome sequencing study and may therefore be considered rigorous.

Methods are overall described with enough detail to replicate although specific references to some of the general methodology used has not been included.For example:
Line 86: Concentration of RNA obtained and methodology for cDNA library construction.
Line 87-89: Indicate software used for clustering, correction, consensus read and redundancy removal.
Line 93: A reference for Transdecoder. In this regard, provide software version numbers for all software used.
Line 95: Provide references for all relevant databases used.
Line 104-108: Provide references for software and databases.

Validity of the findings

The results are standard for this type of study and overall robust and statistically sound.
The conclusions are well stated and founded and support the results.

Additional comments

Figure 1: Y-axis legend should be number.

---

## Round 0.2 · Minor Revisions

Please address the remaining issues indicated by the reviewer and amend your manuscript accordingly.

Reviewer 1 ·

Basic reporting

The overall flow and structure of the manuscript are clear and concise. Tables and figures look good.

Experimental design

This manuscript provides the full-length transcriptome data of A. flabellulatum. Albeit the current manuscript reporting is merely descriptive in nature, this work serves as important baseline comparison reference for future transcriptomic studies of this species. The full-length transcriptome sequencing dan data analyses are rather standard. Additionally, the authors validated the transcriptome data using qPCR, which provided further support to the study. I noticed that there is an inconsistency in the Method, i.e., line 85-88. Did the authors pooled the three replicates of each illumination into one (line 86), or did they extract the RNA of each replicate, as noted in line 92.

Validity of the findings

no comment.

Additional comments

Please add more to the first paragraph of the Introduction. In addition, kindly rephrase the two sentences (line 44-47), 'play an important role' can be found in both first and second sentences.

Reviewer 2 ·

Basic reporting

No comment

Experimental design

No comment

Validity of the findings

No comment

---

## Round 0.3 · accepted · Accept

The remaining critiques are addressed and the manuscript is amended accordingly. Therefore, the revised version is acceptable now.